# The impact of biologic agents on cardiovascular risk factors in patients with rheumatoid arthritis: A meta analysis

Xiaodong Jia[1]☺, Zheming Yang[1,2]☺, Jiayin Li[1,2], Zhu Mei[1,2], Lihui Jia[1]*, Chenghui Yan [1]*

1 State Key Laboratory of Frigid Zone Cardiovascular Diseases (SKLFZCD), Cardiovascular Research Institute and Department of Cardiology, General Hospital of Northern Theater Command, Shenyang, Liaoning, China, 2 College of Medicine and Biological Information Engineering, Northeastern University, Shenyang, Liaoning, China

☺ These authors contributed equally to this work.
* 418211786@qq.com (LJ); yanch1029@163.com (CY)

## Abstract

### Objective

The purpose of the study is to evaluate the effects of biologic therapy on cardiovascular risk factors in rheumatoid arthritis patients to determine its clinical efficacy.

### Methods

Relevant literature was systematically searched in PubMed, Embase, and Cochrane Library databases. Meta-analysis was conducted using standardized mean differences (SMDs) and 95% confidence intervals (CIs) to evaluate cardiovascular risk factors and atherosclerosis. Heterogeneity, sensitivity analysis, and publication bias were assessed. Statistical significance was set at $P<0.05$.

### Results

The meta-analysis revealed that biologic treatment in RA patients was associated with decreased high-density lipoprotein cholesterol (HDL-C) levels compared to controls (MD: -0.10, 95% CI: [-0.14, -0.05], $P<0.0001$). Subgroup analysis based on treatment duration showed heterogeneity and a potential decrease in total cholesterol levels after 12 months of treatment (MD = -0.03, 95% CI [-0.21, -0.15], $P = 0.76$). Biologic therapy significantly reduced triglyceride levels compared to controls (MD = -0.23, 95% CI [-0.37, -0.09], $P = 0.001$), as observed in subgroup analysis. Moreover, biologics effectively decreased low-density lipoprotein cholesterol (LDL-C) levels (MD: -0.10, 95% CI: [-0.14, -0.05], $P<0.0001$). However, biologic treatment was associated with increased inner carotid artery thickness (MD: 0.05, 95% CI: [0.03, 0.07], $P<0.0001$), indicating potential adverse effects on cardiovascular health. No significant effect on pulse wave velocity (PWV) was observed (MD: -0.23, 95% CI: [-0.80, 0.34], $P = 0.43$, I2 = 0%, $P = 0.55$).

**Data Availability Statement:** All relevant data are within the paper and its Supporting information files.

**Funding:** This study was supported by the National Key Research and Development Program of China (2022YFC2503500 and 2022YFC2503504) and the National Natural Science Foundation of China (82070300, 82270300, 32071116 and 82170297). The funders had no role in study design, data collection and analysis, decision to publish, or preparation of the manuscript.

**Competing interests:** The authors have declared that no competing interests exist.

## Conclusion

Biologic agents may improve lipid profiles in RA patients but could also have adverse effects on cardiovascular health. Further research is needed to comprehensively understand the impact of biologic therapy on lipid metabolism and cardiovascular outcomes in RA patients.

## Systematic review registration

https://www.crd.york.ac.uk/PROSPERO/, CRD42024504911.

## Introduction

Rheumatoid arthritis (RA) is a chronic autoimmune disease characterized by inflammation of the synovium, leading to joint destruction and disability, it affects approximately 0.5–1.0% of the population worldwide [1–3]. RA not only impacts the musculoskeletal system but also increases the risk of developing cardiovascular disease (CVD), which is a major cause of morbidity and mortality in RA patients [4]. The association between RA and CVD has been well-established, with studies consistently demonstrating an increased risk of cardiovascular events in patients with RA compared to the general population [5–7]. Traditionally, this increased risk has been attributed to the traditional cardiovascular risk factors such as hypertension, dyslipidemia, smoking, obesity, and diabetes, which are more common among RA patients [8–10]. However, recent evidence suggests that chronic inflammation plays a crucial role in the development of CVD in RA [11].

Over the past few decades, the introduction of biologic agents has revolutionized the treatment of RA [12]. Biologic agents, including tumor necrosis factor-alpha (TNF-α) inhibitors, interleukin-6 (IL-6) receptor antagonists, and B-cell depleting agents, target specific components of the immune system and effectively suppress inflammation in RA [13]. These medications have shown remarkable efficacy in controlling symptoms, improving joint function, and halting radiographic progression in RA patients [14]. However, their impact on cardiovascular risk remains a topic of debate.

Some studies have suggested that biologic agents may have a beneficial effect on cardiovascular outcomes in RA patients [15, 16]. The potent anti-inflammatory properties of these agents may lead to reduced atherosclerosis progression, improved endothelial function, and decreased C-reactive protein levels, thereby attenuating the risk of CVD. On the other hand, there is emerging evidence indicating that certain biologic agents, particularly TNF-α inhibitors, may have neutral or even detrimental effects on cardiovascular risk in RA [17–19].

Given the contradictory findings in the literature, a comprehensive and up-to-date analysis of available evidence is warranted to evaluate the impact of biologic agents on cardiovascular risk in RA patients. We will comprehensively search the literature, identify relevant studies, extract data, and perform statistical analyses to estimate the pooled effect size. Furthermore, we will explore potential sources of heterogeneity, perform subgroup analyses, and conduct sensitivity analyses to assess the robustness of our findings.

## Methods

The review proposal was registered with PROSPERO: CRD42024504911 (S1 Checklist).

## Research design and article retrieval strategies

This meta-analysis aimed to comprehensively assess the association between the use of biologics and cardiovascular risk in patients with RA. The study followed the PRISMA guidelines and conducted a thorough search of the PubMed, Embase, and Cochrane Library databases to identify eligible studies. The search was conducted until May 2023, using MeSH terms and keywords related to Rheumatoid Arthritis, such as "Rheumatoid arthritis" and "RA". MeSH terms and keywords related to "biologics" were then employed, including terms like "Biological Products," "biologic therapy," "biologic drug," as well as specific biologic agents such as "dalimumab," "infliximab," "etanercept," and "certolizumab.pegol, golimumab, tocilizumab, abatacept, rituximab, anakinra, sarilumab," among others. Additionally, MeSH terms and keywords related to "cardiovascular risk" were used, including terms like "cardiovascular diseases," "cardiovascular risk," "cardiovascular events," and "atherosclerosis." The search terms for biological agents were also included, such as "tumor necrosis factor inhibitors" (TNFi), "interleukin-6 receptor blockers" (IL-6R blockers), as well as specific agents like rituximab, tacrolimus, tofacitinib, abatacept, adalimumab, anakinra, etanercept, golimumab, infliximab, sarilumab, tocilizumab, ustekinumab, secuk. Furthermore, the reference lists of relevant literature were screened to ensure that no eligible studies were missed.

## Article selection

Inclusion criteria: (1) Subjects: Subjects were RA patients; (2) Interventions: patients with RA who received immunosuppressants or control group who did not receive immunosuppressants; (3) Outcome indicators: total cholesterol, triglyceride, low density lipoprotein, LDL-C, CIMT, PWV and other indicators; (4) All included studies were randomized controlled trials and retrospective or prospective observational studies.

Exclusion criteria: Study type: non-control study, such as case report, summary report, review, etc. Subjects: (1) non-RA patients; (2) Interventions: RA patients treated with non-immunosuppressive agents; (3) Outcome indicators: No cardiovascular events or related indicators were reported.

## Data extraction

Data extraction primarily collected essential information, including the first author, publication year, country, disease, follow-up time, treatment drug, drug dosage, number of participants, age, gender, and study type. It also involved variables related to the study background, such as measures impacting cardiovascular events and measures evaluating atherosclerosis. To ensure accuracy and reliability, two independent researchers performed data extraction. Disputes arising from data extraction were resolved through discussion and negotiation. After data extraction, a thorough comparison and verification process was conducted to ensure the accuracy and completeness of the extracted data.

## Quality assessment

We used the Risk of Bias 2.0 tool provided by the Cochrane Collaboration Network to assess the quality of the included studies. This tool allows for the evaluation of bias across various aspects of each study, including random sequence generation, allocation concealment, blinding methods, completeness of data, and selective reporting. Each assessment criterion was categorized into three levels of risk: low risk, uncertain risk, and high risk. The assessment results were recorded accordingly. For an overall assessment of the study, we determined its overall quality as either low, medium, or high risk based on the risk level identified for each

assessment criterion. By assessing the quality of the included studies, we aim to gain a better understanding of their limitations and biases. This enables us to more accurately evaluate the reliability and generalizability of the results obtained from the meta-analysis.

## Statistical analysis

The statistical analysis was conducted using STATA 14.0 software (Stata Corporation). The combined effect size of the continuous variable was calculated to determine the potential relationship with cardiovascular risk factors and atherosclerosis using SMDS (Standardized Mean Difference) along with their corresponding 95% confidence intervals (CI). To assess the heterogeneity between studies, the $I^2$ statistic was calculated using the chi-square test. If $I^2 > 50\%$ or $P < 0.10$, indicating the presence of significant heterogeneity, a random effects model was employed. Otherwise, a fixed effects model was used. The source of heterogeneity was further explored through a systematic deletion of individual studies and subgroup analysis. Sensitivity analysis was performed to ensure the stability and reliability of the results. Funnel plots were used to detect possible publication bias. A significance level of $P < 0.05$ was considered statistically significant for all analyses conducted.

## Results

### Literature selection process

Relevant keywords and screening criteria were applied to search the PubMed, Embase, and Cochrane Library databases for relevant literature. Based on the title and abstract, initially, 234 articles were selected that were related to the research question. After excluding studies that were not relevant or did not meet the inclusion criteria, 95 studies remained. The full texts of these 95 studies were then examined, and further selection was made based on whether they met the requirements of the study. As a result, 63 studies that did not meet the requirements were excluded. The remaining 32 studies were read in their entirety. Among them, 11 studies lacked data on cardiovascular risk factors, and 9 studies lacked data on atherosclerosis evaluation indices (Fig 1). Finally, a total of 12 articles were included in the final meta-analysis, involving 1164 patients [20–31].

### Features included in the study

Twelve studies were included in the analysis, covering a time span from 2006 to 2022. The sample sizes in these studies varied, ranging from 20 to 232 participants. The age range of the participants was between 45 and 63 years. The studies were conducted in various countries across different regions: Europe (Denmark, United Kingdom, Greece, Russia, Italy, Norway), Asia (Hong Kong), and North America (United States). The follow-up duration in the included studies varied from 3 to 12 months (Table 1).

### Risk of bias assessment

In our research, a comprehensive risk of bias assessment was conducted for the 12 included studies. The results of this assessment are presented (Figs 2 and 3), providing a detailed evaluation of each study's potential sources of bias along with their respective risk of bias scores. By performing such a meticulous risk of bias assessment, we aimed to ensure the credibility and dependability of our research findings. We employed rigorous methods to evaluate the studies and implemented measures to mitigate any potential biases that could have influenced the results. This meticulous approach enhances the overall quality and reliability of our research.

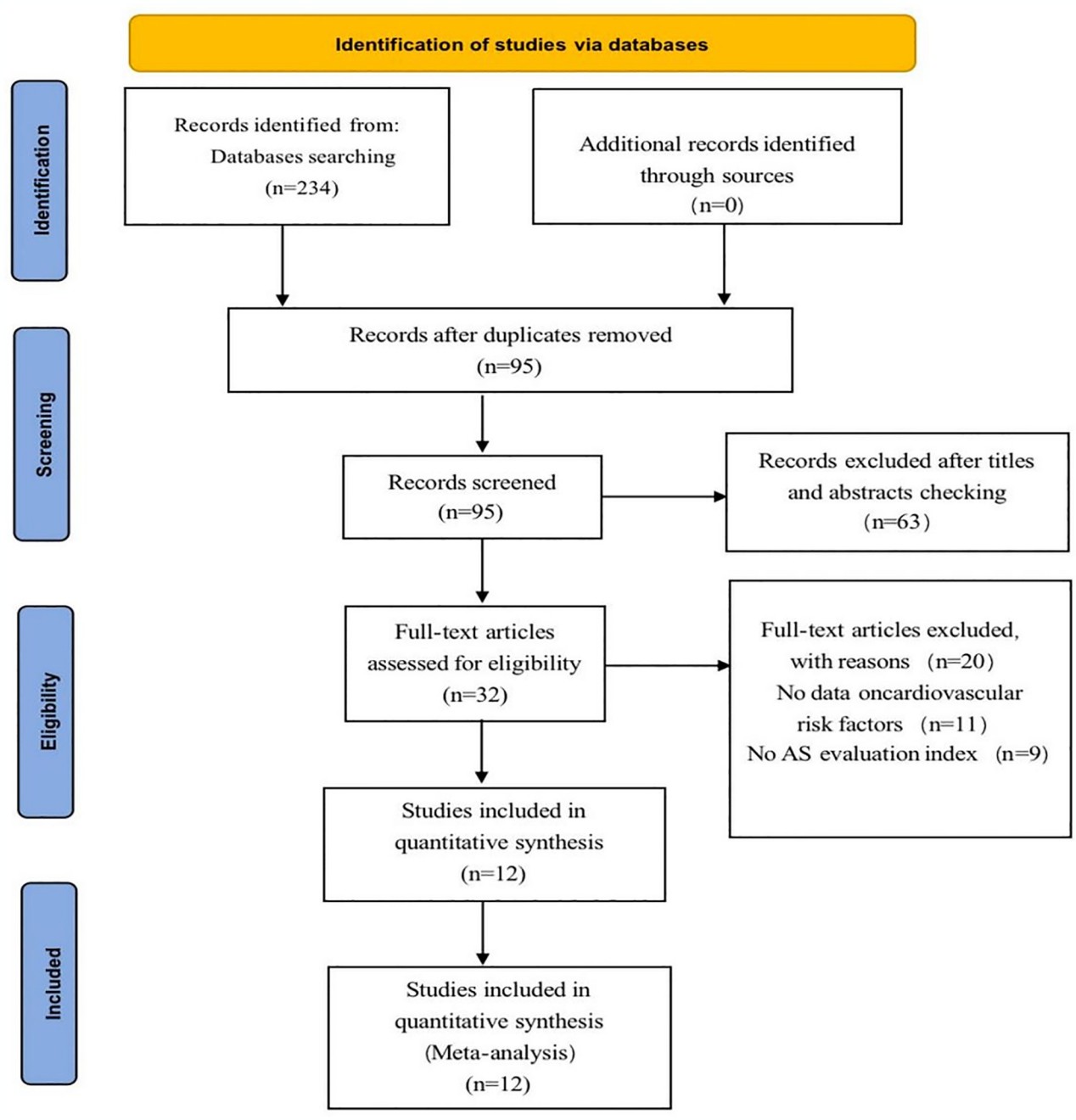

**Fig 1. Inclusion of literature flowchart.**

## Total cholesterol

A total of 12 studies were included in this paper, all of which analyzed the effect of biological agents on total cholesterol levels in RA treatment and were included in the meta-analysis. These studies collectively involved 1043 patients. The forest plot results demonstrated that biologics had a significant effect on reducing total cholesterol levels (mean difference [MD] =

**Table 1. Basic characteristics of research.**

| Author/year | country | disease | Follow-up time | Treatment group/control group | dose | Number of patients | age | Percentage of women |
|---|---|---|---|---|---|---|---|---|
| Mašić D/2021 [20] | Denmark | RA | 12month | ADA+MTX/POB+MTX | 40mg/kg | 174 | 63 | 78% |
| Mäki-Petäjä/2006 [21] | Britain | RA | 12month | TNF-α/placebo | 4mg/kg | 219 | 57 | 59% |
| Wong/2009 [22] | Britain | RA | 12month | TNF-α/placebo | 3mg/kg | 26 | 49 | 84% |
| Papamichail /2022 [23] | Greece | RA | 6month | Rituximab | 4mg/kg | 80 | 55 | 75% |
| Novikova/2016 [24] | Russia | RA | 6month | Rituximab | 22mg/kg | 55 | 50 | 100% |
| O'Neill/2017 [25] | Britain | RA | 13month | MTX+IFX/MTX+POB | 5mg/kg | 36 | 58 | 55% |
| Tam/2012 [26] | Hong Kong | RA | 6month | FX+MTX/MTX | 10mg/kg | 40 | 53 | 85% |
| Hsue/2014 [27] | United States | RA | 6month | Rituximab | 3mg/kg | 95 | 53 | 54% |
| McInnes/2015 [28] | Britain | RA | 6month | TCZ+MTX/placebo MTX | 6mg/kg | 132 | 57 | 52% |
| Ferrante/2009 [29] | Italy | RA | 3month | TNFαi/MTX | 15–20mg | 232 | 45 | 67% |
| Cardillo/2006 [30] | Italy | RA | 12month | Infliximab | | 20 | 45 | 62% |
| Angel/2012 [31] | Norway | RA | 12month | TNF-α | | 55 | 52 | 65% |

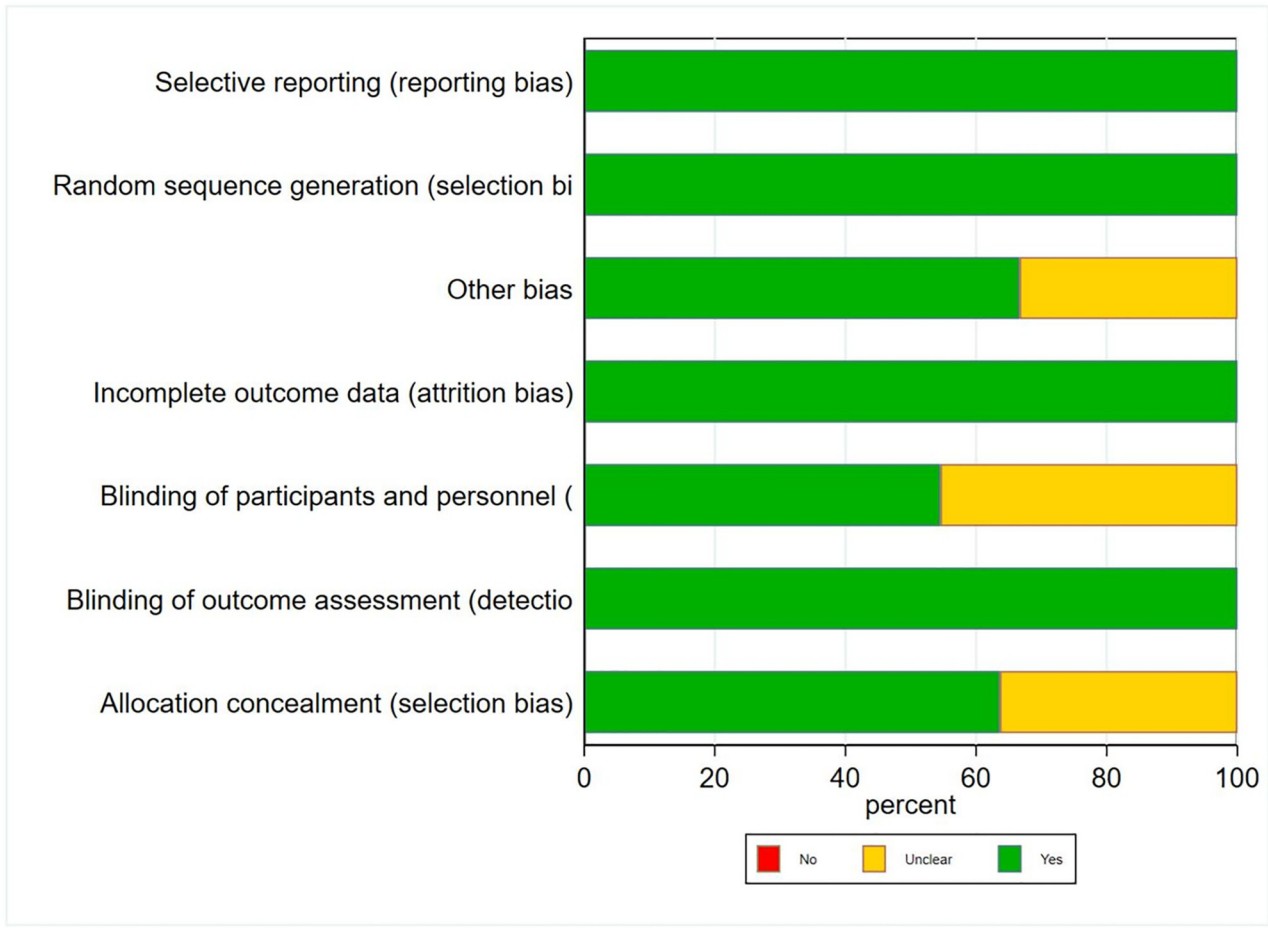

**Fig 2. Review authors judgments about each risk of bias item presented as percentages across all included studies.**

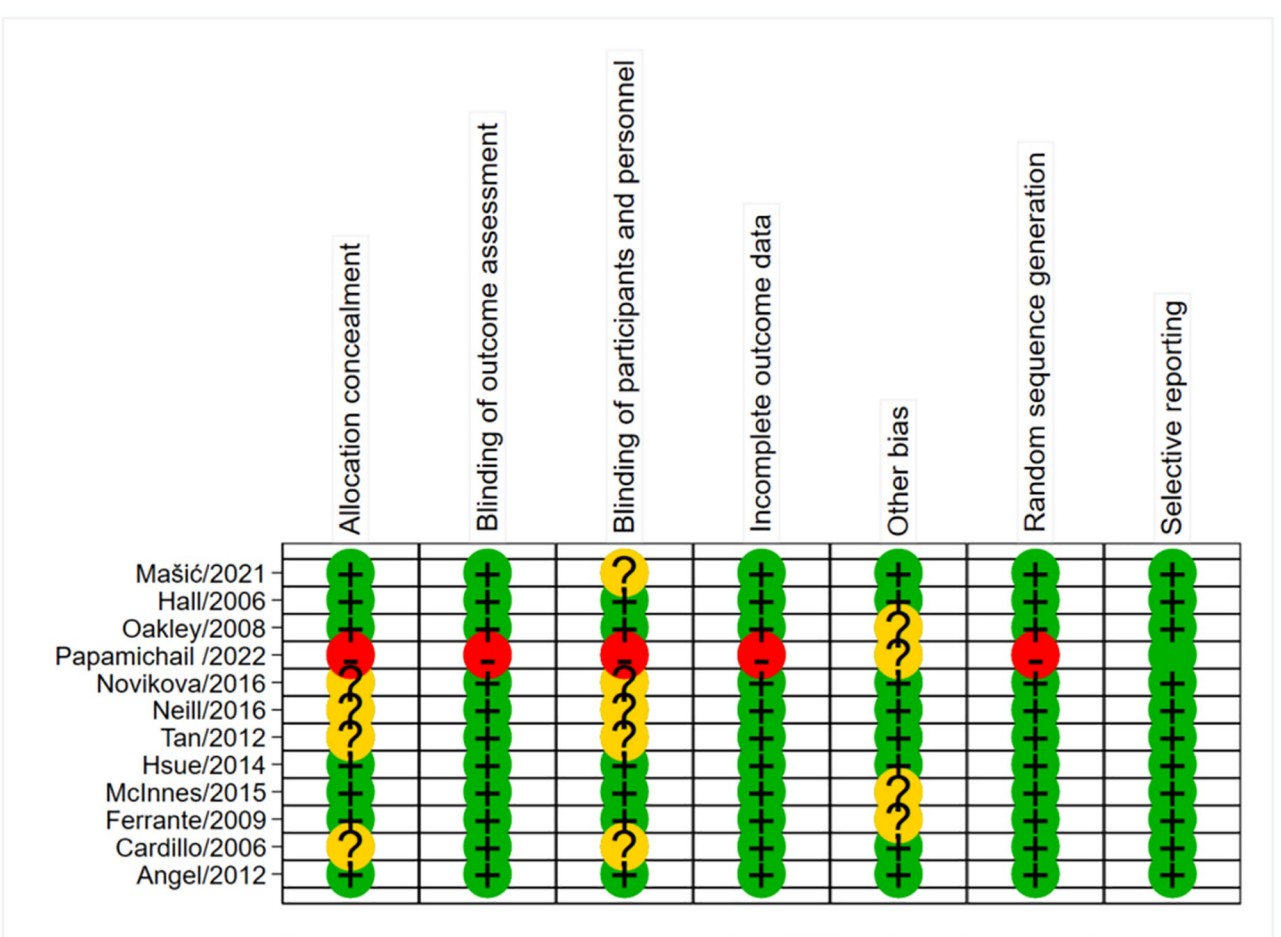

**Fig 3. Reviewer' judgments about each risk of bias item for each included study.**

-0.27, 95% confidence interval [CI: -0.41, -0.13], $P<0.001$). The chi-square test indicated significant heterogeneity among the studies ($I^2 = 81\%$, $P<0.0001$).

Subsequently, a sensitivity analysis was conducted by systematically removing one study at a time. After excluding the study by Novikova et al. (2016), the effect of biologic-based treatment on total cholesterol became non-significant (MD = 0.02, 95% CI: [-0.02,0.18], $P = 0.79$), and the heterogeneity decreased from 81% to 1% (S1 Fig). This suggests that the study by Novikova et al. may have contributed to the observed heterogeneity.

Furthermore, the analysis was stratified based on drug treatment duration, different treatment approaches, study country, and study design type. In the subgroup analysis of drug treatment duration, there was substantial heterogeneity between the two groups, indicating that treatment duration significantly influenced the meta-analysis results ($I^2 = 81\%$, $P<0.001$). When the treatment duration exceeded 12 months, there was no heterogeneity observed, and the results of six studies with treatment durations over 12 months were combined using a fixed-effect model. The pooled effect size showed a trend towards reducing total cholesterol levels in the biologic treatment group (MD = -0.03, 95% CI: [-0.21, -0.15], $P = 0.76$), (z = 0.30, $P = 0.76$). On the other hand, heterogeneity was present when the treatment duration was less than 12 months ($I^2 = 77\%$, $P<0.005$) (Fig 4). In the subgroup analysis of different drug

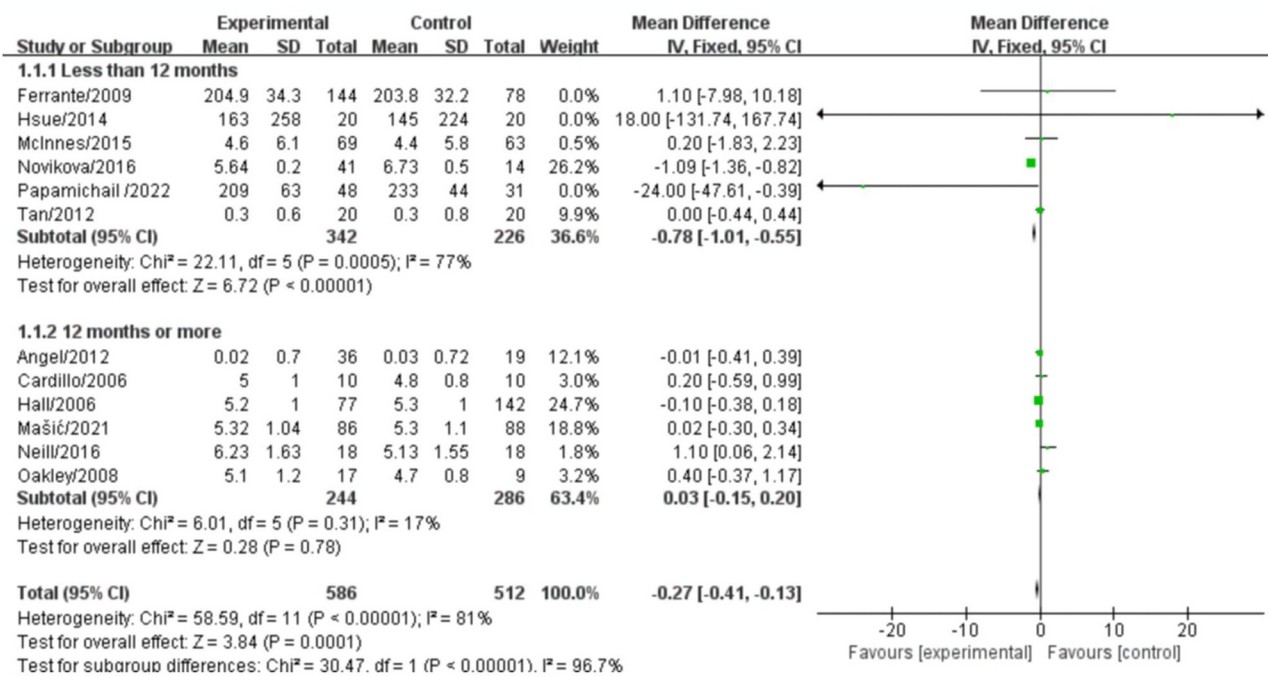

**Fig 4. Forest plot of the effect of biologic therapy versus conventional therapy on total cholesterol levels stratified by treatment duration.**

treatments, no heterogeneity was observed, and the pooled effect size indicated a trend towards a decrease in total cholesterol levels in the biologics with DMARDs treatment group, but it did not reach statistical significance (S2 Fig).

## Triglyceride

A total of 12 studies were included in this paper, all of which analyzed the effects of biologic agents on triglyceride levels in RA treatment. These studies collectively involved 1043 patients. The results showed that the effect of biologics on triglycerides in patients with rheumatic diseases was not statistically significant (mean difference [MD] = -0.06, 95% confidence interval [CI: -0.16, 0.03], $P$ = 0.2), as indicated (S3 Fig).

Further analysis was conducted by stratifying the data based on drug treatment duration, different treatment approaches, study country, and study design type. In the subgroup analysis of drug treatment duration, there was moderate heterogeneity observed between the two groups ($I^2$ = 60%, $P$ = 0.004), indicating that treatment duration significantly influenced the meta-analysis results. When the treatment duration was less than 12 months, there was minimal heterogeneity, and the results of six studies with treatment durations less than 12 months were combined. The fixed-effect model was used to combine the effect size, which demonstrated a significant reduction in triglyceride levels in the biologic treatment group (MD = -0.18, 95% CI [-0.31, -0.05], $P$ = 0.008). Specifically, a significant reduction in triglyceride levels was observed in the biologic treatment group when the treatment duration was less than 6 months. Additionally, heterogeneity was observed after 12 months of treatment ($I^2$ = 64%, $P$ = 0.02) (Fig 5).

When stratified by different treatment approaches, biologic treatment of RA was found to effectively reduce triglyceride levels when using biologics monotherapy (MD = -0.16, 95% CI [-0.28, -0.04], $P$ = 0.01) (Fig 6).

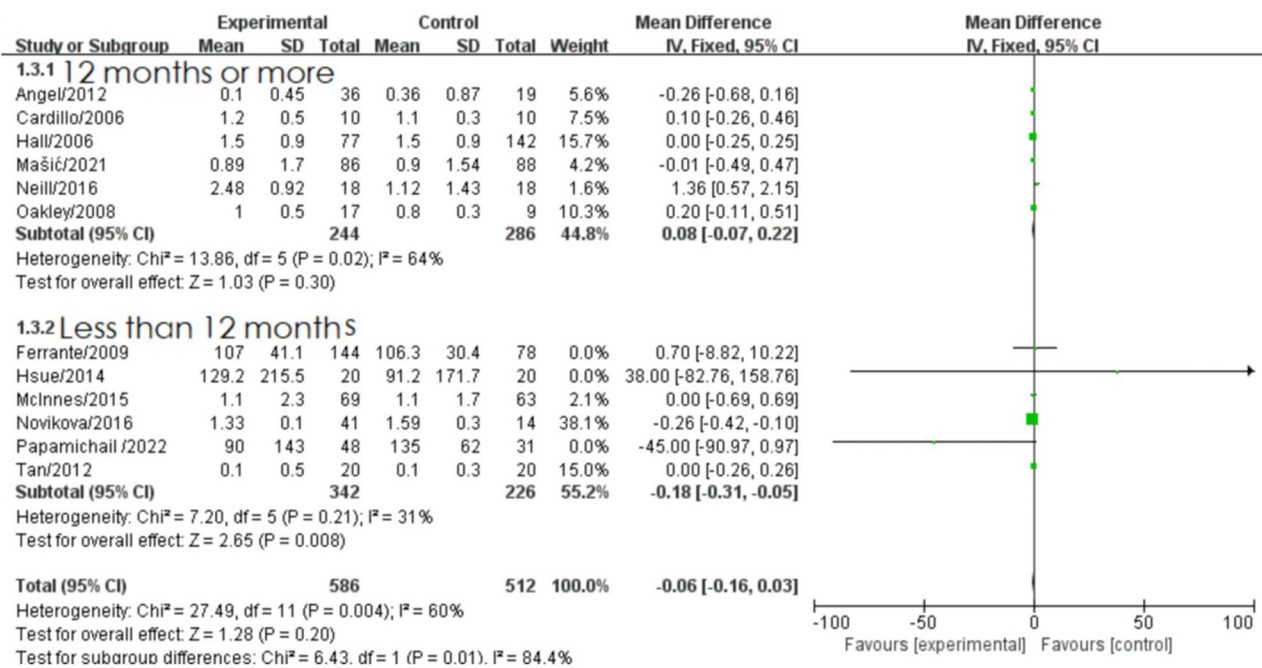

**Fig 5. Forest plot of the effect of biologic therapy versus conventional therapy on triglyceride levels stratified by treatment duration.**

## Low-density lipoprotein

Eleven studies involving 1023 patients were conducted to evaluate the effect of biologics on low-density lipoprotein (LDL) indices. The final meta-analysis results showed that the test results for LDL with biologics were not statistically significant (mean difference [MD]: -0.08,

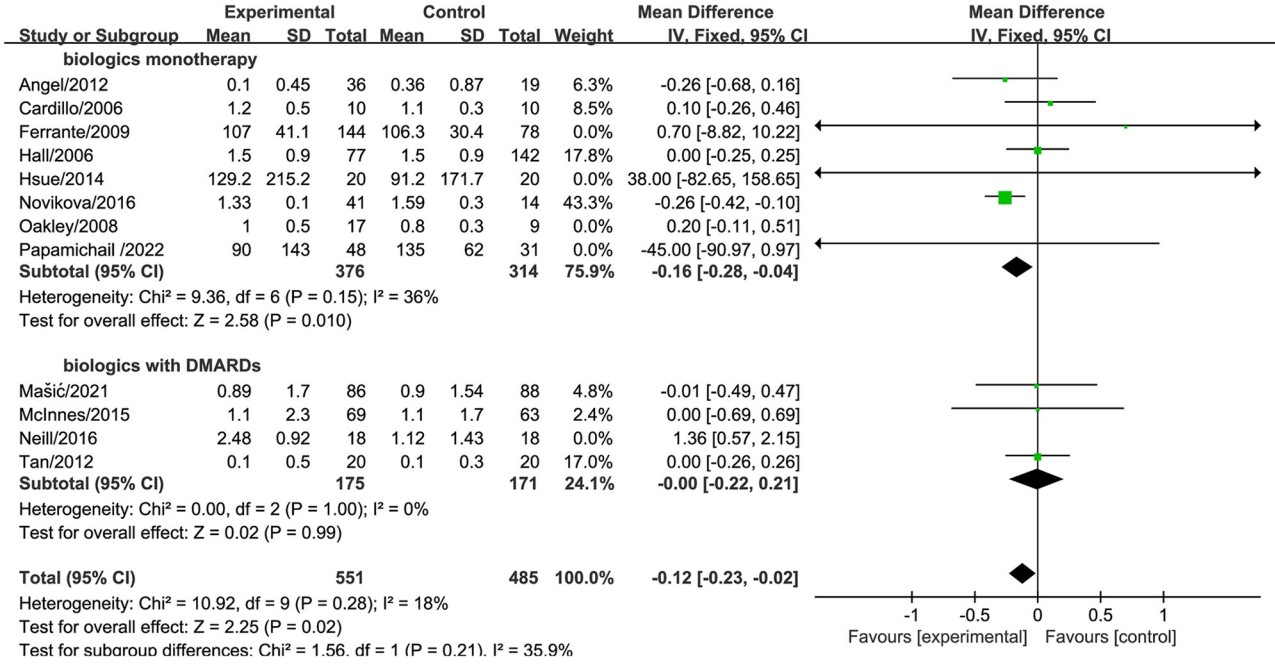

**Fig 6. Forest plot of the effect of biologic therapy versus conventional therapy on triglyceride levels stratified by different drug treatments.**

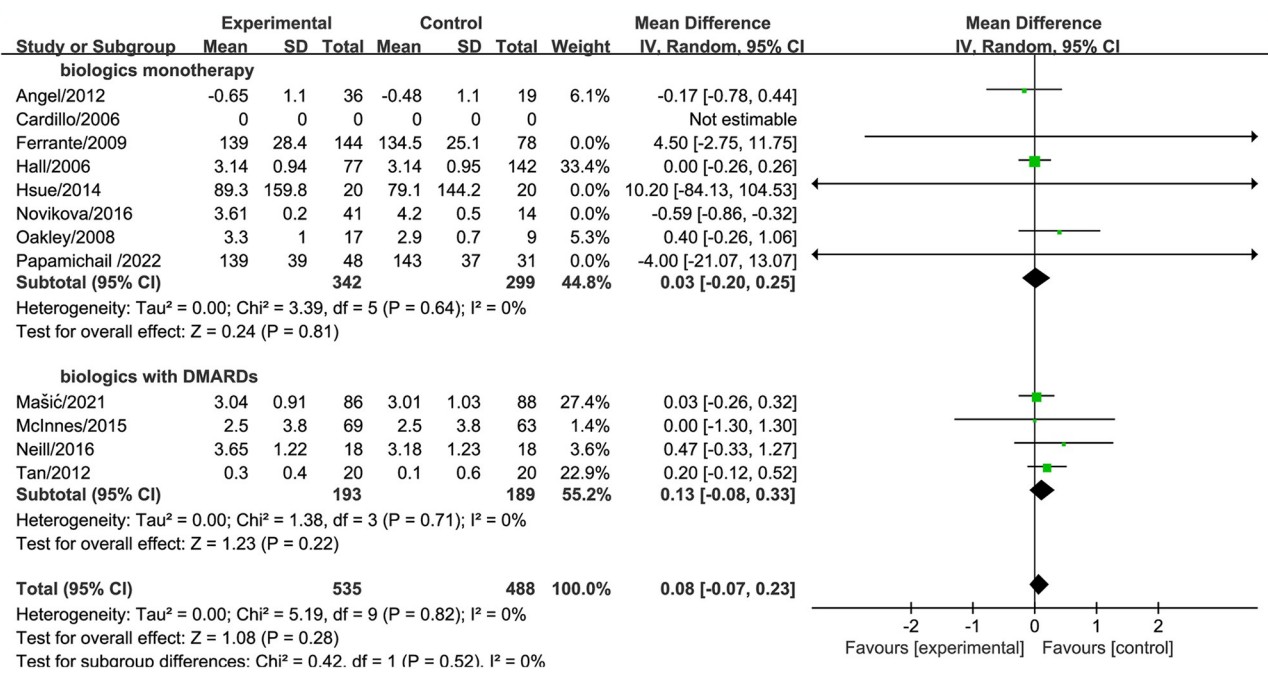

**Fig 7. Forest plot of the effect of biologic therapy versus conventional therapy on low-density lipoprotein levels stratified by treatment duration.**

95% confidence interval [CI: -0.21–0.05, *P* = 0.24), and there was significant heterogeneity observed. After removing the study (Novikova/2016), the combined effect size was recalculated (MD: 0.08, 95% CI: -0.07–0.23, *P* = 0.28), and the heterogeneity decreased from 57% to 0% (S4 Fig). Sensitivity analysis indicated that the article may be the source of heterogeneity.

The subgroup analyses based on different treatment methods and treatment duration did not yield statistically significant results (Figs 7 and 8).

## HDL-C

Twelve studies involving 1043 patients were conducted to evaluate the effect of biologics on HDL cholesterol (HDL-C) levels. The meta-analysis results from these 12 studies indicated a significant difference in HDL-C levels between the biologic-treated RA group and the control group. Specifically, the level of HDL-C in the biologic-treated group was lower than that in the control group (mean difference [MD]: -0.10, 95% confidence interval [CI: -0.14, -0.05], *P*<0.0001). Heterogeneity among the studies was low (*P* = 0.06, I2 = 42%), and a fixed-effects model was used to analyze the data (Fig 9).

## Medial carotid intima thickness

Among the 12 literature included in this study, three studies reported the effect of biologic agents on carotid intima-medial thickness (CIMT) in RA treatment, involving a total of 160 participants. The meta-analysis results indicated a significant difference in CIMT between the biologic treatment group and the control group. Specifically, the biologic treatment group had a higher CIMT compared to the control group (mean difference [MD]: 0.05, 95% confidence interval [CI: 0.03, 0.07], *P*<0.0001). There was no significant heterogeneity observed ($I^2$ = 0%, *P* = 0.39), and a fixed-effects model was used to analyze the data (Fig 10).

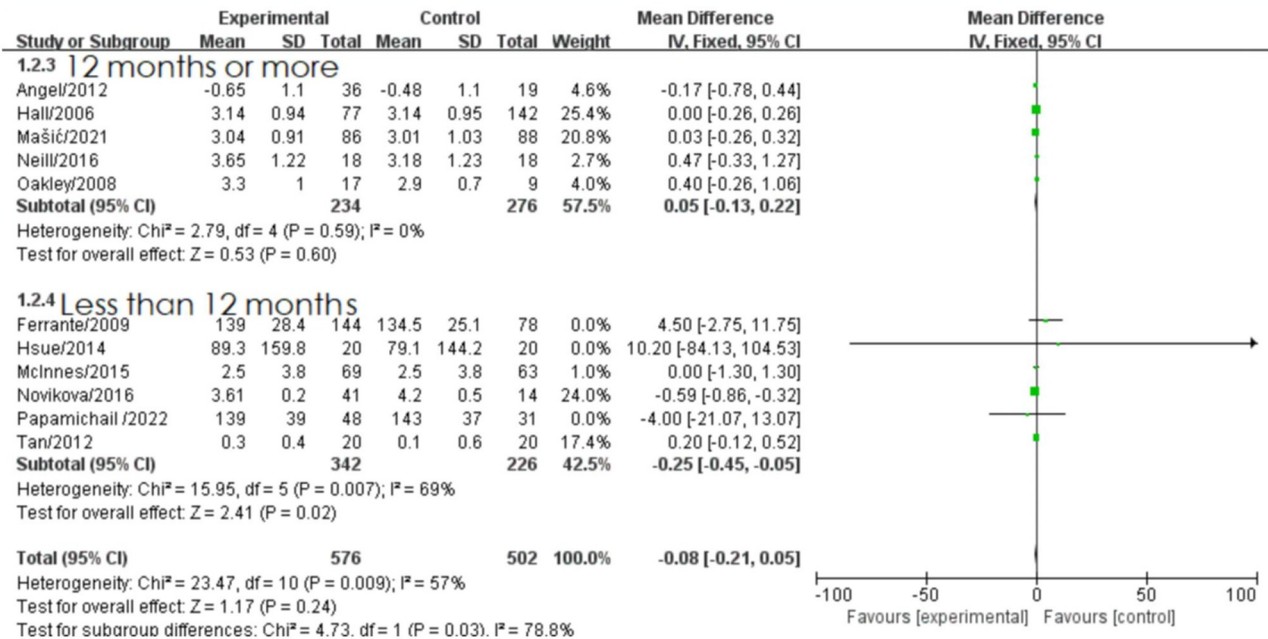

**Fig 8. Forest plot of the effect of biologic therapy versus conventional therapy on low-density lipoprotein levels stratified by different drug treatments.**

## PWV

Among the 12 literature included in this study, four studies reported the effect of biologic agents on pulse wave velocity (PWV) in RA treatment. The meta-analysis results indicated that the effect of biologics on PWV was not statistically significant (mean difference [MD]: -0.23, 95% confidence interval [CI: -0.80, 0.34], $P = 0.43$). There was no heterogeneity observed ($I^2 = 0\%$, $P = 0.55$), and a fixed-effects model was used to analyze the data (Fig 11).

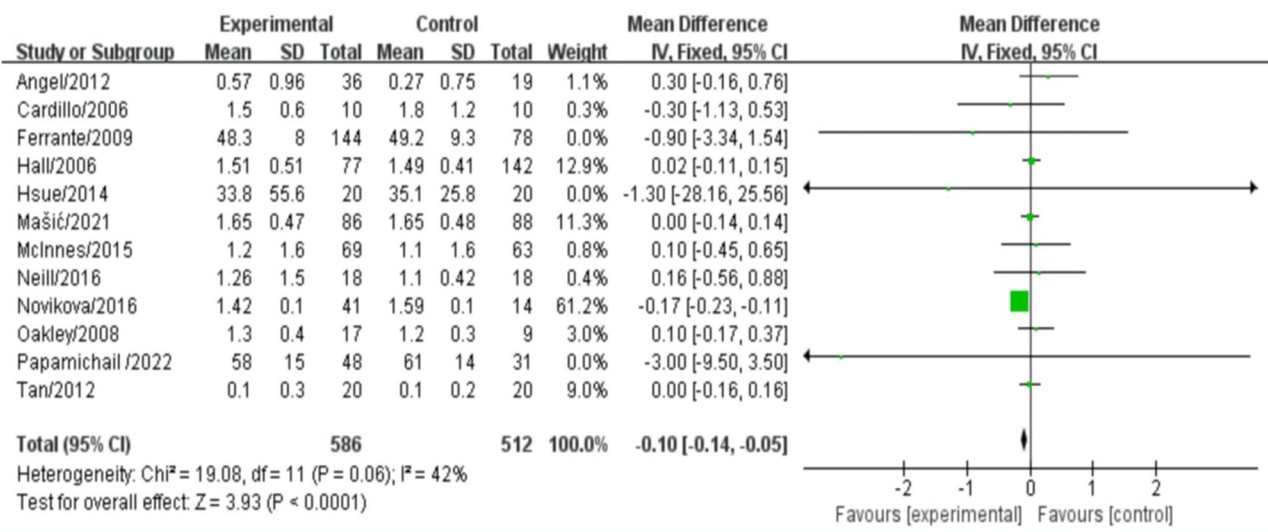

**Fig 9. Meta-analysis results for indexes related to HDL-C.**

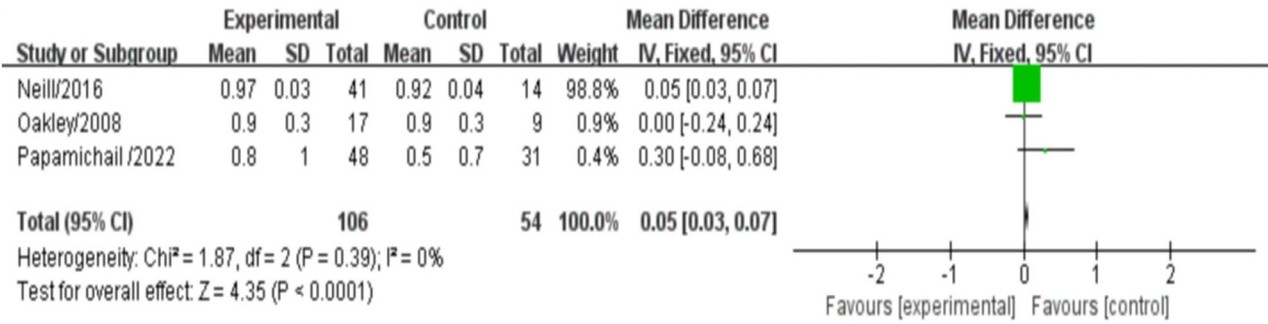

**Fig 10. Meta-analysis results for indexes related to medial carotid intima thickness.**

## Sensitivity analysis

To assess the stability of this meta-analysis, sensitivity analysis was conducted by recalculating the combined data after excluding each study one by one. The results showed no significant changes in the combined HR estimates, indicating that the findings were robust and stable (S5A–S5F Fig).

## Assessment of reporting bias

For total cholesterol, triglycerides, LDL-C, HDL-C, PWV, and CIMT, the effects of biologics and their corresponding 95% CI were combined and analyzed using a funnel plot test to assess publication bias (S6A–S6F Fig). The funnel plots for total cholesterol, triglycerides, LDL-C, HDL-C, PWV, and CIMT exhibited a symmetrical distribution, suggesting no evidence of publication bias.

## Discussion

The results of our meta-analysis indicate that the use of biologic agents in patients with RA may have an impact on lipid metabolism. Specifically, we observed a lower level of HDL-D in the biological-treated group compared to the control group (MD: -0.10, 95%CI: [-0.14, -0.05], $P<0.0001$). Total cholesterol levels showed a trend of reduction in the biological-treated group (MD = -0.03, 95%CI[-0.21, -0.15], $P = 0.76$), particularly when the duration of treatment exceeded 12 months, although it did not reach statistical significance. On the other hand, the use of biologics was associated with a significant reduction in triglyceride (MD = -0.23, 95%CI

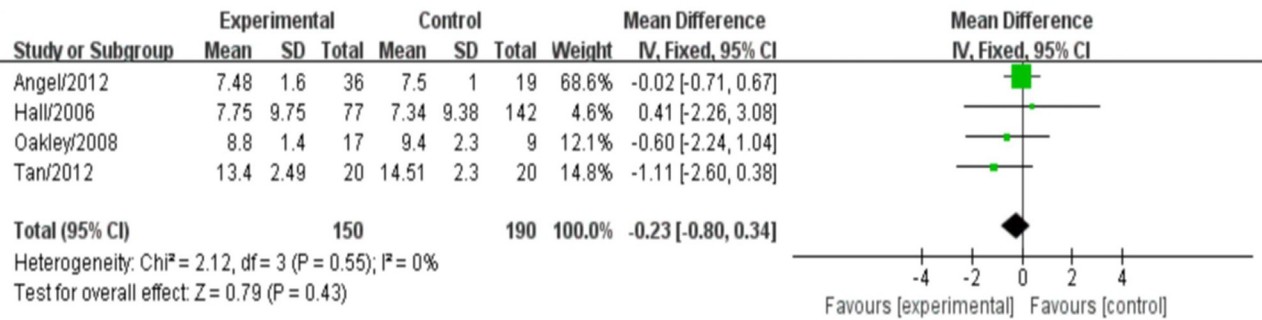

**Fig 11. Meta-analysis results for indexes related to PWV.**

[-0.37, -0.09], *P* = 0.001) and LDL-C levels (MD:0.10, 95%CI:[-0.14, -0.05], *P*<0.0001). These findings suggest that biologic agents may influence lipid profiles in RA patients, which could have implications for cardiovascular risk. Dyslipidemia is a known risk factor for cardiovascular disease (CVD), and maintaining optimal lipid levels is important for reducing the risk of CVD events. The observed reduction in triglyceride and LDL-C levels among patients receiving biologic treatment may contribute to a favorable lipid profile and potentially lower their cardiovascular risk.

Our results are consistent with previous studies that have shown the beneficial effects of biologic agents on lipid profiles in patients with RA. A 2012 meta-analysis also reported a reduction in total cholesterol and LDL-C levels with biologic therapy in RA patients. However, it did not report any significant effect on HDL-C and triglyceride levels, as well as other markers of atherosclerosis such as carotid intima-media thickness and pulse wave velocity [32].

Our meta-analysis results revealed an interesting finding regarding carotid intima-media thickness (CIMT) in patients receiving biologic treatment for RA. We observed that CIMT was higher in the biologic treatment group compared to the control group, and this difference was statistically significant (MD: 0.05, 95%CI: [0.03,0.07], *P*<0.0001). This finding suggests that biologic therapy in RA patients may be associated with increased CIMT, which is an established marker of subclinical atherosclerosis and a predictor of cardiovascular events [33]. The thickening of the carotid intima-media layer is indicative of early atherosclerotic changes and can be considered an important surrogate marker for cardiovascular risk assessment [34].

Interestingly, our finding contrasts with the results of another study that reported a reduction in medial carotid intima thickness with biologic therapy in RA patients. However, it should be noted that this previous study did not find a significant effect on lipid levels [16]. The discrepancy in findings between these studies suggests that the relationship between biologic agents, CIMT, and lipid metabolism in RA patients is complex and may involve multiple factors. Our stratified analyses based on duration of treatment and across different countries revealed a high degree of heterogeneity. This suggests that treatment duration and geographic factors may contribute to the varying effects of biologics on CIMT. It is important to consider these factors when interpreting the results and designing future studies.

While our study provides valuable insights into the association between lipid levels and cardiovascular risk in patients with RA, it is important to acknowledge certain limitations that should be considered when interpreting the results. One of the primary limitations of our study is that although we focus on lipid levels as cardiovascular risk factors in RA, it is important to note that there are other significant risk factors that should be taken into account. Factors such as the use of aspirin, statins, antihypertensive drugs, smoking status, presence of diabetes, familiarity, and obesity play a substantial role in determining the overall cardiovascular risk in these patients. Future studies should aim to comprehensively assess these factors to provide a more accurate evaluation of cardiovascular risk in RA. Furthermore, the observed differences in intima-media thickness (IMT) between patients on biologic therapy and control groups may be influenced by the baseline disease severity and overall cardiovascular risk profile of the patients. Patients not receiving biologic therapy might have milder disease activity, potentially leading to a lower cardiovascular risk compared to those on biologic agents. This confounding factor should be carefully considered and addressed in future analyses to ensure an accurate assessment of the impact of biologic therapy on cardiovascular risk. It is also worth noting that chronic inflammation associated with RA can lead to alterations in lipid metabolism, resulting in changes in lipid levels that may not align with the expected cardiovascular risk profile. While biologic therapy targeting inflammation may improve lipid profiles, the complex interplay between inflammation and lipid metabolism, known as the "lipid paradox," may limit the translation of improved lipid profiles into reduced cardiovascular risk. Future

studies should further investigate this interrelationship to better understand how inflammation affects lipid metabolism and its implications for assessing cardiovascular risk in RA. Lastly, the short-term nature of our study limits our ability to capture any potential long-term changes or trends related to the effects of biologic therapy on cardiovascular risk factors in RA patients. Longitudinal studies are essential for obtaining a more comprehensive understanding of the long-term impact of biologic agents on cardiovascular health in this population. Additionally, incorporating a broader range of traditional and non-traditional risk factors is crucial for a more holistic evaluation of cardiovascular outcomes in RA patients.

In summary, the impact of biologic therapy on lipid profiles appears to be influenced by the duration of treatment. Within less than 12 months of treatment, significant reductions in triglyceride levels were observed in the biologically-treated group. However, there was no statistically significant trend in decreasing total cholesterol and LDL-C levels over a longer treatment period. Furthermore, biologic therapy was associated with lower HDL-C levels and increased CIMT, suggesting a potential adverse effect on cardiovascular health. However, the effect on PWV did not reach statistical significance. It is important to note that several analyses exhibited a high degree of heterogeneity, indicating the presence of varying factors among the included studies. Consequently, further research is needed to validate these findings and explore potential underlying contributors to the observed changes. To gain a more comprehensive understanding of the impact of biologic therapy on lipid metabolism and cardiovascular health, future studies should consider evaluating additional factors beyond lipid levels. These factors include inflammatory markers, endothelial dysfunction, and arterial stiffness, as they play significant roles in cardiovascular risk assessment. Moreover, long-term studies examining clinical outcomes, such as cardiovascular events and prognosis, are warranted to determine the overall effects of biologic therapy.

## Supporting information

**S1 Checklist. PRISMA checklist.**
(DOCX)

**S1 Table. Study characteristics assessing the quality of the included studies.**
(XLSX)

**S1 Fig. Effect of biological agents on total cholesterol outcomes forest results.**
(TIF)

**S2 Fig. Forest plot of the effect of biologic therapy versus conventional therapy on total cholesterol levels stratified by different drug treatments.**
(TIF)

**S3 Fig. Effect of biologics on triglyceride outcomes of RA forest results.**
(TIF)

**S4 Fig. Effect of biologics on low density lipoprotein (LDL).**
(TIF)

**S5 Fig. Sensitivity analysis.** A: Total cholesterol B: triglyceride C: low density lipoprotein D: HDL-C E: medial carotid intima thickness F: PWV.
(TIF)

**S6 Fig. Assessment of reporting bias.** A: Total cholesterol B: triglyceride C: low density lipoprotein D: HDL-C E: medial carotid intima thickness F: PWV.
(TIF)

## Acknowledgments

All authors read and approved the final manuscript. We are grateful to all the researchers involved in this study.

## Author Contributions

**Conceptualization:** Xiaodong Jia, Lihui Jia, Chenghui Yan.

**Formal analysis:** Xiaodong Jia, Zheming Yang, Zhu Mei.

**Methodology:** Xiaodong Jia, Lihui Jia, Chenghui Yan.

**Software:** Xiaodong Jia.

**Supervision:** Lihui Jia, Chenghui Yan.

**Writing – original draft:** Xiaodong Jia.

**Writing – review & editing:** Zheming Yang, Jiayin Li, Lihui Jia, Chenghui Yan.

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
