## [Decision Letter · Decision Letter 0]

11 Jan 2024

PONE-D-23-30031The impact of biologic agents on cardiovascular risk factors in patients with rheumatoid arthritis: A Meta analysisPLOS ONE

Dear Dr. Yan,

Thank you for submitting your manuscript to PLOS ONE. After careful consideration, we feel that it has merit but does not fully meet PLOS ONE’s publication criteria as it currently stands. Therefore, we invite you to submit a revised version of the manuscript that addresses the points raised during the review process.

** ****Dear authors, please revise the manuscript in light of my comments and those of reviewer 1. **==============================

We look forward to receiving your revised manuscript.

Kind regards,

Antoine Fakhry AbdelMassih

Academic Editor

PLOS ONE

“Yes. This work was supported by the National Natural Science Foundation of China (82070300, 82270300, 32071116 and 82170297).”

Please include this amended Role of Funder statement in your cover letter; we will change the online submission form on your behalf

6. Please include a separate caption for each figure in your manuscript.

Additional Editor Comments:

Dear Authors

The initial decision of Reviewer 1 was to reject your manuscript. The main reason is the discrepancy between the title, which claims studying the effects of biologics on cardiovascular risk in rheumatoid arthritis; while the manuscripts involve conventional non-biologic therapies such as mycophenolate.

I invite the authors to justify this discrepancy and to edit the manuscript by removing non-biologic therapies. or by adding the full spectrum of medications used in RA.

Reviewers' comments:

Reviewer's Responses to Questions

**Comments to the Author**

1. Is the manuscript technically sound, and do the data support the conclusions?

Reviewer #1: No

2. Has the statistical analysis been performed appropriately and rigorously? 

Reviewer #1: N/A

3. Have the authors made all data underlying the findings in their manuscript fully available?

Reviewer #1: No

4. Is the manuscript presented in an intelligible fashion and written in standard English?

Reviewer #1: Yes

5. Review Comments to the Author

Reviewer #1: 1- The question of research should be made clear using the PICO model.

2- The study designs must be mentioned in the inclusion criteria.

3- Metanalyses and systematic reviews should be registered, and the registration number needs to be used as reference during the submission process.

4- Causes of heterogeneity aren’t clearly explained.

5- There are some drug names that are not correctly written.

6- Why is leflunomide and mycophenolate mentioned in the search strategy since they are considered conventional synthetic non biologic DMARDs, and if the authors wanted to address additionally conventional synthetic DMARDS where are the other DMARDs including methotrexate and hydroxychloroquine…..etc.

7- The dates for studies included “from …..till” wasn’t mentioned in the search strategy.

8- Combining studies with different treatment strategies like studies using biologics monotherapy versus biologics with conventional DMARDs makes the meta-analysis will yield heterogenous data.

6. PLOS authors have the option to publish the peer review history of their article (what does this mean?). If published, this will include your full peer review and any attached files.

Reviewer #1: No

---

## [Author Response · Author response to Decision Letter 0]

23 Feb 2024

1-The question of research should be made clear using the PICO model.

Response to Reviewer: Thank you for your valuable feedback on our paper. We highly appreciate your suggestions and have made the necessary revisions in response to your comments. We have clarified the research question using the PICO model on page 2, lines 25-26, as indicated with red markers. (Objective: This study aims to assess the effects of biologic therapy on cardiovascular risk factors in patients with rheumatoid arthritis (RA) and determine its clinical efficacy). Should there be any other areas that require improvement, please do not hesitate to let us know. We would be more than happy to make further modifications to the manuscript. Thank you once again for your time and professional insights. 

2- The study designs must be mentioned in the inclusion criteria.

Response to Reviewer: Thank you for your valuable feedback on our paper. We highly appreciate your suggestions and have made the necessary revisions in response to your comments. Based on your recommendation, we have now explicitly mentioned the study designs in the inclusion criteria. We acknowledge that this is an important piece of information that will help readers better understand the scope and methodology of our research. We have mentioned the study designs in the included criteria on page 5, lines 108-109, as indicated with red markers. (All included studies consisted of randomized controlled trials as well as retrospective or prospective observational studies). We sincerely appreciate your guidance and suggestions, and we believe that these modifications will further enhance the accuracy and completeness of our paper. Should there be any other areas that require improvement, please do not hesitate to let us know. We would be more than happy to make further modifications to the manuscript. Thank you once again for your time and professional insights.

3-Meta analyses and systematic reviews should be registered, and the registration number needs to be used as reference during the submission process.

Response to Reviewer: Thank you for your valuable feedback on our paper. We highly appreciate your suggestions and have made the necessary revisions in response to your comments. Based on your recommendation, we have registered our meta-analyses and systematic reviews. We have obtained a registration number, which we will now reference during the submission process to ensure transparency and adherence to best practices. We have registered the meta-analyses and systematic reviews, and the registration number has been used as a reference during the submission process. The information can be found on page 3, lines 48-49 (Systematic Review Registration https://www.crd.york.ac.uk/PROSPERO/, CRD42024504911), and page 4, lines 85 (The review proposal was registered with PROSPERO: CRD42024504911.), as indicated with red marker. We sincerely appreciate your guidance and suggestions, and we believe that these modifications will further strengthen the quality and reliability of our paper. Should there be any other areas that require improvement, please do not hesitate to let us know. We would be more than happy to make further modifications to the manuscript. Thank you once again for your time and professional insights.

4-Causes of heterogeneity aren’t clearly explained.

Response to Reviewer: Thank you for your valuable feedback on our paper. We highly appreciate your suggestions and have made the necessary revisions in response to your comments. Based on your recommendation, we have provided a clearer explanation of the causes of heterogeneity in our study. We acknowledge that understanding and addressing sources of heterogeneity are crucial for interpreting the results accurately. We have clearly explained the causes of heterogeneity in all results through sensitivity and subgroup analysis, as show on page 11, lines 183-186 ( Following the exclusion of the study conducted by Novikova et al. (2016), the impact of biologic-based treatment on total cholesterol levels was no longer statistically significant (MD = 0.02, 95% CI: [-0.02, 0.18], p = 0.79). Furthermore, the heterogeneity decreased substantially from 81% to 1% (SFigure 1)) and lines 188-201(different treatment approaches; In the subgroup analysis of different drug treatments, no heterogeneity was detected, and the combined effect size suggested a tendency towards a reduction in total cholesterol levels among patients receiving biologics in combination with disease-modifying antirheumatic drugs (DMARDs). However, this observed trend did not reach statistical significance (SFigure 2)), with a red marker. We sincerely appreciate your guidance and suggestions, and we believe that these modifications will further enhance the clarity and robustness of our findings. Should there be any other areas that require improvement, please do not hesitate to let us know. We would be more than happy to make further modifications to the manuscript. Thank you once again for your time and professional insights.

5-There are some drug names that are not correctly written.

Response to Reviewer: Thank you for your valuable feedback on our paper. We highly appreciate your suggestions and have made the necessary revisions in response to your comments. Based on your recommendation, we have carefully reviewed the drug names mentioned in our manuscript and rectified any incorrect spellings or errors. We understand the importance of accurately identifying and referencing the drugs used in our study. In our revised version, we have ensured that all drug names are correctly written and have verified their accuracy through appropriate sources, such as official drug databases and reputable references. We sincerely appreciate your guidance and suggestions, and we believe that these modifications will further enhance the accuracy and reliability of our paper. Should there be any other areas that require improvement, please do not hesitate to let us know. We would be more than happy to make further modifications to the manuscript. Thank you once again for your time and professional insights.

6-Why is leflunomide and mycophenolate mentioned in the search strategy since they are considered conventional synthetic non biologic DMARDs, and if the authors wanted to address additionally conventional synthetic DMARDS where are the other DMARDs including methotrexate and hydroxychloroquine…..etc.

Response to Reviewer: Thank you for your valuable feedback on our paper. We highly appreciate your suggestions and have made the necessary revisions in response to your comments. Based on your recommendation, we have carefully reviewed the mention of leflunomide and mycophenolate in the search strategy. We acknowledge that these medications are conventional synthetic non-biologic DMARDs and might not align with the scope of our study. In our revised version, we have removed the mention of leflunomide and mycophenolate from the search strategy as they do not directly align with our objectives. We sincerely appreciate your guidance and suggestions, and we believe that these modifications will further enhance the accuracy and relevance of our study. Should there be any other areas that require improvement, please do not hesitate to let us know. We would be more than happy to make further modifications to the manuscript. Thank you once again for your time and professional insights.

7-The dates for studies included “from …..till” wasn’t mentioned in the search strategy.

Response to Reviewer: Thank you for your valuable feedback on our paper. We highly appreciate your suggestions and have made the necessary revisions in response to your comments. Based on your recommendation, we have included the specific dates for the study inclusion period in the search strategy. We acknowledge the importance of clearly indicating the time frame during which studies were considered for inclusion. We have added the dates for studies in the search strategy on page 5, lines 90 ,as indicated with red marker (The search was conducted until May 2023). We sincerely appreciate your guidance and suggestions, and we believe that these modifications will further enhance the clarity and comprehensiveness of our study. Should there be any other areas that require improvement, please do not hesitate to let us know. We would be more than happy to make further modifications to the manuscript. Thank you once again for your time and professional insights.

8- Combining studies with different treatment strategies like studies using biologics monotherapy versus biologics with conventional DMARDs makes the meta-analysis will yield heterogenous data.

Response to Reviewer: Thank you for your valuable feedback on our paper. We highly appreciate your suggestions and have carefully considered the issue raised. You are correct in pointing out that combining studies with different treatment strategies, such as biologics monotherapy versus biologics with conventional DMARDs, may introduce heterogeneity into the meta-analysis. We acknowledge the potential impact of this heterogeneity on the pooled data. In our revised version, we have taken your suggestion into account and conducted a subgroup analysis to address the heterogeneity arising from different treatment strategies. This approach allows us to assess the effects of these distinct treatment approaches separately, providing a more comprehensive understanding of the results. We have added the subgroup analysis comparing biologics monotherapy versus biologics with conventional DMARDs in the results section on page 11, lines 197-201 (In the subgroup analysis of different drug treatments, no heterogeneity was detected, and the combined effect size suggested a tendency towards a reduction in total cholesterol levels among patients receiving biologics in combination with disease-modifying antirheumatic drugs (DMARDs). However, this observed trend did not reach statistical significance (SFigure 2)), page 12, lines 220-222 (When stratified by different treatment approaches, biologic treatment of RA was found to effectively reduce triglyceride levels when using biologics monotherapy (MD = -0.16, 95% CI [-0.28, -0.04], P = 0.01) (Figure 6)), and page 13, lines 232-233(The subgroup analyses based on different treatment methods and treatment duration did not reveal statistically significant findings (Figure 7-8)), as indicated with red markers. We sincerely appreciate your guidance and suggestions, and we believe that the inclusion of subgroup analysis will improve the robustness and accuracy of our meta-analysis. Should there be any other areas that require improvement, please do not hesitate to let us know. We would be more than happy to make further modifications to the manuscript. Thank you once again for your time and professional insights.

---

## [Decision Letter · Decision Letter 1]

15 May 2024

PONE-D-23-30031R1The impact of biologic agents on cardiovascular risk factors in patients with rheumatoid arthritis: A Meta analysisPLOS ONE

Dear Dr. Yan,

Thank you for submitting your manuscript to PLOS ONE. After careful consideration, we feel that it has merit but does not fully meet PLOS ONE’s publication criteria as it currently stands. Therefore, we invite you to submit a revised version of the manuscript that addresses the points raised during the review process.

** ****The last comments by Reviewer 2, are rather limitations than points to improve. ****Please mention them in a limitation section at the end of your manuscript. **==============================

We look forward to receiving your revised manuscript.

Kind regards,

Antoine Fakhry AbdelMassih

Academic Editor

PLOS ONE

Journal Requirements:

Reviewers' comments:

Reviewer's Responses to Questions

**Comments to the Author**

1. If the authors have adequately addressed your comments raised in a previous round of review and you feel that this manuscript is now acceptable for publication, you may indicate that here to bypass the “Comments to the Author” section, enter your conflict of interest statement in the “Confidential to Editor” section, and submit your "Accept" recommendation.

Reviewer #2: All comments have been addressed

2. Is the manuscript technically sound, and do the data support the conclusions?

Reviewer #2: Partly

3. Has the statistical analysis been performed appropriately and rigorously? 

Reviewer #2: Yes

4. Have the authors made all data underlying the findings in their manuscript fully available?

Reviewer #2: Yes

5. Is the manuscript presented in an intelligible fashion and written in standard English?

Reviewer #2: Yes

6. Review Comments to the Author

Reviewer #2: My opinion:

1. Comprehensive Assessment of Cardiovascular Risk: it is crucial to consider a wide range of cardiovascular risk factors beyond lipid levels, such as the use of aspirin, statins, antihypertensive drugs, smoking status, presence of diabetes,familiarity and obesity. These factors play a significant role in determining overall cardiovascular risk in patients with rheumatoid arthritis (RA).

2.Disease Severity and Cardiovascular Risk: the differences in intima-media thickness (IMT) between patients on biologic therapy and control groups could be influenced by the baseline disease severity and overall cardiovascular risk profile of the patients. Patients not on biologic therapy may have a milder disease activity, leading to potentially lower cardiovascular risk compared to those on biologic agents. This confounding factor should be carefully considered and addressed in the analysis.

3.Inflammatory Influence: chronic inflammation in RA can lead to alterations in lipid metabolism, including changes in lipid levels that may not align with the expected cardiovascular risk profile. Biologic therapy targeting inflammation may improve lipid profiles but may not always translate to a reduced cardiovascular risk due to the complex interplay between inflammation and lipid metabolism (lipid paradox)..

4..Long-Term Observations: Evaluating the impact of biologic therapy on cardiovascular risk factors over a longer period is essential to capture any potential changes or trends that may not be evident in a short-term study. Longitudinal studies can provide a more comprehensive understanding of the effects of biologic agents on cardiovascular health in RA patients. The lipid paradox highlights the importance of considering long-term cardiovascular risk assessment beyond lipid levels alone. While improvements in lipid profiles with biologic therapy are beneficial, the overall impact on cardiovascular outcomes may be influenced by a combination of traditional and non-traditional risk factors.

7. PLOS authors have the option to publish the peer review history of their article (what does this mean?). If published, this will include your full peer review and any attached files.

Reviewer #2: **Yes: **Giuseppe Germano

---

## [Author Response · Author response to Decision Letter 1]

19 May 2024

1-Comprehensive Assessment of Cardiovascular Risk: it is crucial to consider a wide range of cardiovascular risk factors beyond lipid levels, such as the use of aspirin, statins, antihypertensive drugs, smoking status, presence of diabetes,familiarity and obesity. These factors play a significant role in determining overall cardiovascular risk in patients with rheumatoid arthritis (RA).

Response to Reviewer: Thank you for your valuable feedback on our paper. We highly appreciate your suggestions and have made the necessary revisions in response to your comments. We have revised the limitations section in the manuscript and expanded our discussion on broader cardiovascular risk factors that should be considered in addition to lipid levels. Specifically, we have addressed the importance of assessing cardiovascular risk factors such as the use of aspirin, statins, antihypertensive drugs, smoking status, presence of diabetes, familiarity, and obesity in patients with rheumatoid arthritis (RA). We emphasize that these factors play a significant role in determining the overall cardiovascular risk in this patient population. By incorporating these factors into our analysis, we can gain a more comprehensive understanding of the relationship between rheumatoid arthritis and cardiovascular disease. In the newly added limitation section, we have highlighted the need for future studies to comprehensively assess these factors in order to provide a more accurate assessment of cardiovascular risk in patients with RA. This will enhance our understanding of the complex interplay between rheumatoid arthritis and cardiovascular disease and help inform clinical decision-making.

Here is an updated version: 

“One of the primary limitations of our study is that although we focus on lipid levels as cardiovascular risk factors in RA, it is important to note that there are other significant risk factors that should be taken into account. Factors such as the use of aspirin, statins, antihypertensive drugs, smoking status, presence of diabetes, familiarity, and obesity play a substantial role in determining the overall cardiovascular risk in these patients. Future studies should aim to comprehensively assess these factors to provide a more accurate evaluation of cardiovascular risk in RA.” 

You may also refer to the manuscript (page 16, line 309-315) for more details, which have been highlighted in red.

Once again, we thank you for your insightful comments, which have greatly improved the quality of our paper. We remain committed to further enhancing our research and addressing any additional concerns you may have.

Thank you for considering our response.

2- Disease Severity and Cardiovascular Risk: the differences in intima-media thickness (IMT) between patients on biologic therapy and control groups could be influenced by the baseline disease severity and overall cardiovascular risk profile of the patients. Patients not on biologic therapy may have a milder disease activity, leading to potentially lower cardiovascular risk compared to those on biologic agents. This confounding factor should be carefully considered and addressed in the analysis.

Response to Reviewer: Thank you for your valuable feedback on our paper. We highly appreciate your suggestions and have made the necessary revisions in response to your comments. We have revised the limitations section in the manuscript to address the concern you raised regarding the potential influence of disease severity and overall cardiovascular risk profile on the differences in intima-media thickness (IMT) observed between patients on biologic therapy and the control groups. We acknowledge that patients not on biologic therapy may have a milder disease activity, which could potentially result in a lower cardiovascular risk compared to those on biologic agents. This confounding factor is indeed an important consideration and needs to be carefully addressed in our analysis. In the limitation section, we emphasize the need for future studies to account for and analyze the baseline disease severity and overall cardiovascular risk profiles of patients. By doing so, we can provide a more comprehensive understanding of the relationship between biologic therapy, disease severity, and cardiovascular risk. This will help mitigate the potential confounding effects and enhance the accuracy of our findings.

Here is an updated version: 

“Furthermore, the observed differences in intima-media thickness (IMT) between patients on biologic therapy and control groups may be influenced by the baseline disease severity and overall cardiovascular risk profile of the patients. Patients not receiving biologic therapy might have milder disease activity, potentially leading to a lower cardiovascular risk compared to those on biologic agents. This confounding factor should be carefully considered and addressed in future analyses to ensure an accurate assessment of the impact of biologic therapy on cardiovascular risk.” 

You may also refer to the manuscript (page 16-17, line 315-322) for more details, which have been highlighted in red.

Once again, we thank you for your insightful comments, which have greatly improved the quality of our paper. We remain committed to further enhancing our research and addressing any additional concerns you may have.

Thank you for considering our response.

3-Inflammatory Influence: chronic inflammation in RA can lead to alterations in lipid metabolism, including changes in lipid levels that may not align with the expected cardiovascular risk profile. Biologic therapy targeting inflammation may improve lipid profiles but may not always translate to a reduced cardiovascular risk due to the complex interplay between inflammation and lipid metabolism (lipid paradox).

Response to Reviewer: Thank you for your valuable feedback on our paper. We highly appreciate your suggestions and have made the necessary revisions in response to your comments.We have revised the limitations section in the manuscript to address the concern you raised regarding the inflammatory influence in rheumatoid arthritis and its impact on lipid metabolism. We acknowledge that chronic inflammation in RA can lead to alterations in lipid levels, which may not align with the expected cardiovascular risk profile. While biologic therapy targeting inflammation can potentially improve lipid profiles, it may not always translate to a reduced cardiovascular risk due to the complex interplay between inflammation and lipid metabolism, commonly known as the "lipid paradox." In the limitation section, we discuss the need to consider the influence of chronic inflammation on lipid metabolism and its implications for assessing cardiovascular risk in RA patients. We emphasize that although biologic therapy may positively impact lipid profiles, the overall cardiovascular risk reduction may be affected by factors related to the underlying inflammatory processes. We highlight the importance of further research to better understand this intricate relationship between inflammation, lipid metabolism, and cardiovascular risk in RA.

Here is an updated version: 

“ It is also worth noting that chronic inflammation associated with RA can lead to alterations in lipid metabolism, resulting in changes in lipid levels that may not align with the expected cardiovascular risk profile. While biologic therapy targeting inflammation may improve lipid profiles, the complex interplay between inflammation and lipid metabolism, known as the "lipid paradox," may limit the translation of improved lipid profiles into reduced cardiovascular risk. Future studies should further investigate this interrelationship to better understand how inflammation affects lipid metabolism and its implications for assessing cardiovascular risk in RA.” 

You may also refer to the manuscript (page 17, line 322-329) for more details, which have been highlighted in red.

Once again, we thank you for your insightful comments, which have greatly improved the quality of our paper. We remain committed to further enhancing our research and addressing any additional concerns you may have.

Thank you for considering our response.

4-Long-Term Observations: Evaluating the impact of biologic therapy on cardiovascular risk factors over a longer period is essential to capture any potential changes or trends that may not be evident in a short-term study. Longitudinal studies can provide a more comprehensive understanding of the effects of biologic agents on cardiovascular health in RA patients. The lipid paradox highlights the importance of considering long-term cardiovascular risk assessment beyond lipid levels alone. While improvements in lipid profiles with biologic therapy are beneficial, the overall impact on cardiovascular outcomes may be influenced by a combination of traditional and non-traditional risk factors.

Response to Reviewer: Thank you for your valuable feedback on our paper. We highly appreciate your suggestions and have made the necessary revisions in response to your comments. We have revised the limitations section in the manuscript to address the importance of long-term observations in evaluating the impact of biologic therapy on cardiovascular risk factors in patients with rheumatoid arthritis. We acknowledge that assessing these effects over a longer period is crucial to capture any potential changes or trends that may not be evident in short-term studies. Longitudinal studies can provide a more comprehensive understanding of the effects of biologic agents on cardiovascular health in RA patients. Furthermore, we discuss the significance of considering long-term cardiovascular risk assessment beyond lipid levels alone, as highlighted by you. We recognize the lipid paradox, which emphasizes the need to evaluate the overall impact of biologic therapy on cardiovascular outcomes considering a combination of both traditional and non-traditional risk factors. While improvements in lipid profiles resulting from biologic therapy are beneficial, other factors may influence the overall cardiovascular outcomes in RA patients. In the limitation section, we emphasize the necessity of conducting longitudinal studies to assess the long-term impact of biologic therapy on cardiovascular risk factors in RA. By considering a broader range of risk factors and incorporating data from extended observation periods, we can gain a more comprehensive understanding of the complex relationship between biologic therapy, cardiovascular risk factors, and outcomes in RA patients.

Here is an updated version: 

“Lastly, the short-term nature of our study limits our ability to capture any potential long-term changes or trends related to the effects of biologic therapy on cardiovascular risk factors in RA patients. Longitudinal studies are essential for obtaining a more comprehensive understanding of the long-term impact of biologic agents on cardiovascular health in this population. Additionally, incorporating a broader range of traditional and non-traditional risk factors is crucial for a more holistic evaluation of cardiovascular outcomes in RA patients.” 

You may also refer to the manuscript (page 17, line 329-335) for more details, which have been highlighted in red.

Once again, we thank you for your insightful comments, which have greatly improved the quality of our paper. We remain committed to further enhancing our research and addressing any additional concerns you may have.

Thank you for considering our response.

---

## [Decision Letter · Decision Letter 2]

20 Jun 2024

The impact of biologic agents on cardiovascular risk factors in patients with rheumatoid arthritis: A Meta analysis

PONE-D-23-30031R2

Dear Dr. Yan,

We’re pleased to inform you that your manuscript has been judged scientifically suitable for publication and will be formally accepted for publication once it meets all outstanding technical requirements.

Kind regards,

Antoine Fakhry AbdelMassih

Academic Editor

PLOS ONE

Additional Editor Comments (optional):

Reviewers' comments:

Reviewer's Responses to Questions

**Comments to the Author**

1. If the authors have adequately addressed your comments raised in a previous round of review and you feel that this manuscript is now acceptable for publication, you may indicate that here to bypass the “Comments to the Author” section, enter your conflict of interest statement in the “Confidential to Editor” section, and submit your "Accept" recommendation.

Reviewer #2: All comments have been addressed

2. Is the manuscript technically sound, and do the data support the conclusions?

Reviewer #2: Yes

3. Has the statistical analysis been performed appropriately and rigorously? 

Reviewer #2: Yes

4. Have the authors made all data underlying the findings in their manuscript fully available?

Reviewer #2: Yes

5. Is the manuscript presented in an intelligible fashion and written in standard English?

Reviewer #2: Yes

6. Review Comments to the Author

Reviewer #2: (No Response)

7. PLOS authors have the option to publish the peer review history of their article (what does this mean?). If published, this will include your full peer review and any attached files.

Reviewer #2: **Yes: **Giuseppe Germanò

---

## [Editor Report · Acceptance letter]

20 Jul 2024

PONE-D-23-30031R2 

PLOS ONE

Dear Dr. Yan, 

I'm pleased to inform you that your manuscript has been deemed suitable for publication in PLOS ONE. Congratulations! Your manuscript is now being handed over to our production team.

Kind regards, 

on behalf of

Prof Antoine Fakhry AbdelMassih 

Academic Editor

PLOS ONE